# Optical Imaging of Beta-Amyloid Plaques in Alzheimer’s Disease

**DOI:** 10.3390/bios11080255

**Published:** 2021-07-29

**Authors:** Ziyi Luo, Hao Xu, Liwei Liu, Tymish Y. Ohulchanskyy, Junle Qu

**Affiliations:** Center for Biomedical Photonics, College of Physics and Optoelectronic Engineering, Shenzhen University, Shenzhen 518060, China; luoziyi2020@email.szu.edu.cn (Z.L.); hxuhao@szu.edu.cn (H.X.); liulw@szu.edu.cn (L.L.); tyo@szu.edu.cn (T.Y.O.)

**Keywords:** optical imaging, β-amyloid protein (Aβ), Alzheimer’s disease (AD), fluorescence microscopy, nonlinear optical imaging

## Abstract

Alzheimer’s disease (AD) is a multifactorial, irreversible, and incurable neurodegenerative disease. The main pathological feature of AD is the deposition of misfolded β-amyloid protein (Aβ) plaques in the brain. The abnormal accumulation of Aβ plaques leads to the loss of some neuron functions, further causing the neuron entanglement and the corresponding functional damage, which has a great impact on memory and cognitive functions. Hence, studying the accumulation mechanism of Aβ in the brain and its effect on other tissues is of great significance for the early diagnosis of AD. The current clinical studies of Aβ accumulation mainly rely on medical imaging techniques, which have some deficiencies in sensitivity and specificity. Optical imaging has recently become a research hotspot in the medical field and clinical applications, manifesting noninvasiveness, high sensitivity, absence of ionizing radiation, high contrast, and spatial resolution. Moreover, it is now emerging as a promising tool for the diagnosis and study of Aβ buildup. This review focuses on the application of the optical imaging technique for the determination of Aβ plaques in AD research. In addition, recent advances and key operational applications are discussed.

## 1. Introduction

Alzheimer’s disease (AD) is an irreversible neurodegenerative disease caused by a variety of factors [1]. The disease is still currently incurable, being the number three cause of mortality worldwide after cardiovascular diseases and cancer [2,3,4]. According to a recent World Alzheimer’s report [5], a new case of AD appears every 3 s. Moreover, 50 million cases of AD currently exist worldwide and are expected to reach 152 million in 2050. However, the disease still cannot be identified in the early stages and cannot be effectively cured, and its exact mechanism remains unclear. Many patients have missed the precious opportunity for early treatment due to not being timely diagnosed because the incubation period of AD can be as long as 10 to 20 years. Thus, emphasizing that the survival rate and the quality of life of patients can be greatly improved is important if the disease can be detected at early stages and given an effective treatment. Therefore, the development of methods for early AD diagnosis is of the highest significance and value for the treatment and prevention of AD.

A large number of senile plaques composed of Aβ in the brain of patients with AD has been shown [6]. These plaques produce a neurotoxic effect, causing structural destruction and neuronal network function and make the brain microenvironment of patients with AD significantly different from that of the normal brain [7,8,9]. The Aβ theory has been recognized as the most classical theory in AD pathogenesis. A large number of studies have confirmed the central position of the Aβ theory in AD pathogenesis [10]. The mainstream theory believes that the production and deposition of Aβ is the root and fuse of AD, and the development of anti-Aβ therapeutics remains to be a rational approach to AD treatment [11]. Therefore, the study of the deposited Aβ is conducive to further understand AD pathogenesis and the development of diagnostic methods. In vivo Aβ detection would be a feasible way to realize the early AD diagnosis. However, a lack of high sensitivity detection methods still exists.

The imaging techniques for the clinical AD diagnosis currently mainly include computed tomography (CT) [12], magnetic resonance imaging (MRI) [13,14,15,16,17,18], functional magnetic resonance imaging (fMRI) [19], positron emission tomography (PET) [20], single-photon emission computed tomography (SPECT) [21,22], magnetic resonance spectroscopy (MRS) [23], and so on. In addition, MRI, PET, and SPECT have shown effectiveness in Aβ imaging of the AD brain. However, the imaging of brain pathological changes under the microscope is limited by the long acquisition time and restricted spatial resolution. In addition, their high cost and use of radioactive isotopes or ionizing radiation make them difficult to be used as an early screening tool for everyone. Most importantly, subjective factors in diagnosis appear as the imaging interpretation depends on the clinical experience of the physician. Thus, alternative technologies that are relatively cheap and easy to apply have recently attracted increasing attention.

Compared with clinical imaging modalities, optical imaging possesses many advantages (e.g., noninvasiveness, high sensitivity, low cost, high imaging speed, and the ability to carry out three-dimensional imaging with high spatial and temporal resolution), allowing for the detection of biological processes at the cellular or molecular level [24,25,26]. The application of fluorescence microscopy allows for the use of fluorescent probes to label Aβ in histological studies [27,28]. However, traditional microscopy can be prone to axial and lateral interference, which results in images becoming blurry. The emergence of confocal laser scanning microscopy (CLSM) provides a solution to the axial and lateral interference of fluorescence signal and also allows for optical sectioning, which can be used for three-dimensional imaging of thicker samples [29,30]. However, the use of fluorescent probes may cause certain toxic effects to living organisms, restraining the in vivo imaging. In recent years, researchers have devoted not only to the development of many new near-infrared fluorescent probes [31,32] but also label-free imaging [33,34]. Nonlinear optical imaging is widely used in the study of dynamic biological and physicochemical processes in vitro and in vivo due to the advantages of label-free, chemically specific, and high-speed imaging [35,36,37,38,39,40,41]. In recent years, nonlinear optical imaging technology has also been employed in the AD research field. In particular, it was applied to the detection of Aβ and tau aggregates, neurofibrillary tangles, and cerebral amyloid angiopathy; assessment of dendritic spines and accumulation of senile plaques; evaluation of the development of TPPP/p25 aggregates in patients with AD [42,43,44]. This review briefly introduces optical imaging methods and focuses on their application to Aβ plaque imaging (Figure 1).

## 2. Conventional Fluorescence Microscopy Imaging

Fluorescence microscopy (FM) is a very powerful tool in biomedicine, which takes ultraviolet as the light source, the wavelength is short, and the resolution is higher than that of ordinary microscopy. The FM principle scheme is shown in Figure 2. FM uses special dichromatic mirrors that reflected shorter wavelengths of light and transmitted longer wavelengths of light. Thus, only the longer wavelength red light from the object can be seen and not the scattered violet light. In addition, FM is commonly used to study the absorption and transportation of intracellular substances, and the distribution and location of chemical substances. Moreover, it has been widely used in the biomedicine field due to its strong detection ability, minor stimulation to the organism, and multiple staining in vivo.

When FM is used in the AD diagnosis field, although Aβ protein itself does not have strong endogenous fluorescence, the specific luminescence mechanism of the fluorescent probe can be used to realize the visualization of Aβ protein. Moreover, an exogenous fluorescent probe can be used to label and visualize Aβ proteins. Fluorescent dyes used for protein labeling can fluoresce brightly at very low concentrations and are generally nontoxic to the body, allowing for in vivo studies. In 2011, Karonyo-Hamaoui et al. [45] reported that Aβ plaques can be found earlier in the retina than that in the brain and accumulate as the disease progresses. They used curcumin as a fluorescent label to perform in vivo fluorescence imaging of Aβ plaques in the retina of AD-Tg mice by utilizing a Micron II rodent retinal imaging microscope (a comprehensive ophthalmology research platform developed by Phoenix Research Labs specially designed for small experimental animals’ eyes, as shown in Figure 3). Those plaques were not detected in the non-Tg mice. Furthermore, the effect of MOG45D [46], which has been reported to effectively inhibit Aβ plaque load on retinal plaques was evaluated, and the plaque changes were validated using curcumin as a fluorescent probe. In the 2014 Alzheimer’s Society International Conference, Frost et al. [47] used a novel system of neurovision imaging and a technique called retinal amyloid imaging to detect Aβ in the eye. The patient also had PET imaging of the Aβ in the brain. The results showed that Aβ levels in the retina were closely related to those in the brain and confirmed that the retina was also an area of high plaque deposition. Furthermore, in 2018, Tes et al. [48] tested Cy5 and CRANAD-2 (a compound derived from curcumin) as fluorescent probes for observing the growth of plaques in the retina with a fluorescence imaging system. The result shows that CRANAD-2 is a better fluorescent probe for Aβ imaging. Cy5 can colocate with CRANAD-2, which lays a foundation for fluorescence imaging of labeled Aβ.

In the Aβ imaging of the brain in vitro, Ikonomovic et al. [49] recently used the FM to compare the ability of two fluorescence probes (i.e., flutemetamol and PiB) to bind to plaques. The results showed that the plaque load of flutemetamol and PiB are strongly correlated, and both of them correspond with the Aβ immunohistochemistry.

## 3. Confocal Laser Scanning Microscopy Imaging

The excitation light, in general, irradiates on the sample to generate fluorescence, and the fluorescence signal is strongest at the focal point of the lens. However, some light scattered by the excitation light will also irradiate other parts of the sample to produce a fluorescence signal on a nonconjugated surface. This will cause fuzzy and faint fluorescence contributions from outside of the focal plane. In addition, a confocal microscope solves the interference of axial and lateral direction of image because the focal point of the objective lens and pinhole point are conjugate points to each other to obtain high resolution (Figure 4). A pinhole is used to block non-focal plane signals to eliminate focus blurring. Confocal microscopy can filter out signals reaching the detector from out of the focal plane because of the pinhole. This allows for collecting signals from focal planes one by one when changing focus and using the collected images to reconstruct the three-dimensional structure of thick samples. Using confocal microscopy, samples can be scanned and imaged to analyze the three-dimensional spatial structure of cells without damage (Figure 5) [50]. This technique allows for the optimum visualization of the brain structures, producing informative images.

In the study of the relationship between Aβ plaques and astrocytes or microglia, Li [51] investigated the variance in the endocytosis, transport, and degradation mechanisms of β-amyloid monomer and oligomer on astrocytes by CLSM, and found that both sAβ_42_ and oAβ_42_ could enter astrocytes via macrophage. In a later study, Icke et al. [52] used immunohistochemistry and CLSM to not only collect three-dimensional image datasets from AD mouse models but also automatically detect plaques and their associated microglial responses as well as neuronal damage. Preliminary studies demonstrated that plaque-associated microglia clustering is correlated with plaque size. The data showed that plaque-associated microglia had similar neuroprotective effects on small and large plaques in the early stages of the disease. However, this kind of protection decreased in the late stage. This approach will allow for unbiased quantitative assessment of potential neuroprotective effects in preclinical AD models with Aβ pathology.

Photobiomodulation (PBM) [53] is a developing field of biomedical research. PBM uses low-level/intensity/power laser or monochromatic light to regulate biological functions and has been used to treat AD and other neurodegenerative diseases in recent years. Hannah et al. [54] demonstrated that stimulating the eyes of AD mice with a 40-Hz light-emitting diode flash to induce gamma waves eliminated amyloid deposits. Moreover, Singer et al. [55] have also reached the above conclusion observed through the CLSM. These observations indicate that light induction may cause a systemic effect in the brain, which promoted PBM feasibility in AD treatment. However, further study is still needed to confirm whether it will be therapeutic in human AD.

Alsunusi et al. [56] used the CLSM in the study of the Aβ aggregation mechanism to detect Aβ localization in PC12. In each PC12 construct, the Aβ immunoreactivity response was found to begin in the neurons at 12 h and to initiate plaque deposition, which indicated that the Aβ accumulation increased with time. However, more studies are needed to determine the deposition and AD pathogenesis caused by intraneuronal Aβ aggregation.

In addition to the study of the brain section, the retinal amyloid deposits have also been found by CLSM. Furthermore, Chibhabha et al. [57] found that the staining of the curcumin micelles are colocalized with that of the anti-Aβ_42_ antibody, and both can excellently stain the plaque region in the hippocampal sections and retina. This study provides a basis for optical imaging of retinal amyloid plaques through the eye in vivo.

However, the spatial resolution of wide-field/confocal FM was limited by the Abbe/Rayleigh limit of light diffraction and could not distinguish structures below 200 nm. Researchers have also recently adopted some new optical imaging methods to improve the image resolution of Aβ plaques. Moreover, Wang et al. [58] synthesized a series of aggregation-induced emission fluorogens to enhance fluorescence intensity. Both PD-NA and PD-NA-TEG fluorescence probes exhibit excellent binding to Aβ plaques, and the imaging resolution can be improved to <100 nm using the super-resolution fluorescence imaging system, allowing for clearer observation of Aβ deposition. Furthermore, Ni et al. [59] developed a novel imaging approach by overlaying signals extracted from small focal points to form high-resolution images. Transcranial amyloid deposition can be imaged across the entire brain of mice with AD at a 20-μm resolution by using this large field of view multi-focus FM in combination with a near-infrared dye. To further improve the resolution of the three-dimensional imaging system, Prof. Hui Gong’s team from Wuhan National Research Center for Optoelectronics used the self-developed fluorescence micro-optical sectioning tomography system to image the whole brain and completed the first high-resolution three-dimensional reconstruction of hypothalamic–neurohypophysial system. Immunostaining showed that the detection rate of Aβ plaques larger than 10 μm in diameter was 97.71% ± 0.18%. A dataset of Aβ plaque distributed throughout the brain of 5XFAD transgenic mice was obtained with an imaging resolution of 0.32 × 0.32 × 2 μm [60]. This approach will contribute to the comprehensive and efficient study of the pathogenesis and efficacy evaluation of AD.

## 4. Near-Infrared Fluorescence Imaging

Near-infrared fluorescence imaging (NIRF) has been rapidly developed in recent years because of its high sensitivity, noninvasiveness, simple operation, and the ability to avoid the interference of spontaneous fluorescence in biological tissues. NIRF fluorescent imaging mainly includes the development of biocompatible NIR fluorescent dyes and the synthesis of various probes. During conventional fluorescence imaging in the visible range, both excitation and emission are absorbed or scattered by the tissue, resulting in significant attenuation of the detected emission signal. Moreover, the tissue autofluorescence (which is mostly in the visible range) also interferes with the emission signal acquisition. The scattering and absorption of excitation and emission can be effectively reduced and the required lower excitation energy causes less damage to the biological tissue when the wavelength of the probe is located in the near-infrared region (NIR, 650 to 900 nm). The NIRF has become an essential tool for the analysis of biological samples in vitro and in vivo imaging of small animals. Moreover, the use of amyloid-specific excitable fluorescence probes with emission at 600 to 700 nm has the advantage in comparison with conventionally used probes as they are well suited for deep imaging of amyloid plaques in AD mouse brain in vivo [61,62]. Therefore, the use of NIRF probes targeting Aβ plaques in the brain is of great significance for preclinical study, histopathological examination, and construction of AD models [63,64,65]. NIRF fluorescent imaging mainly includes the development of biocompatible NIR fluorescent dyes and the synthesis of various probes. The ideal NIRF fluorescent probe should have the following characteristics: (1) targeting Aβ with high selectivity and high affinity, (2) the emission wavelength is in the near-infrared spectral range, (3) rapid penetration of the blood–brain barrier (BBB) and rapid clearance in the organism, (4) high quantum yield, (5) low affinity with bovine serum albumin (BSA), (6) easy synthesis, and most importantly, (7) binding to Aβ plaques, which should significantly change its fluorescence properties (e.g., fluorescence intensity, fluorescence lifetime, emission wavelength, and quantum yield). In this section, the application of NIR probes in Aβ imaging and some commercial NIR imaging instruments will be described.

Raymond et al. [25,65] compared specific probes for Aβ with emission wavelengths at 630 and 800 nm and showed that the probes with emission wavelengths at 800 nm had a better signal-to-noise ratio and a higher affinity for Aβ, which would improve the current near-infrared amyloid imaging capability and beyond that of AO1987 (with absorption and emission peaks at 650 and 750 nm, which makes it the most efficient probe for Aβ plaque in the brain at that time). Moreover, curcumin is a specific probe for imaging Aβ plaques. However, its application in near-infrared imaging is limited due to its short emission wavelength and limited transmission through the blood–brain barrier. Furthermore, Ran et al. [64] designed a NIRF probe derived from curcumin, CRANAD-2, with emission at 805 nm. The fluorescence intensity was significantly increased by 70 times in the presence of Aβ aggregates compared with the fluorescence intensity observed in phosphate-buffered saline (pH 7.4). In addition, CRANAD-2 was colocalized with standard thioflavin T staining signals in an in vitro brain tissue analysis. In in vivo experiments, the fluorescence intensity of the transgenic group was higher than that of the control group, and the plaques were colocalized with histological staining signals.

Schmidt et al. [66] then developed in 2012 a novel NIRF probe THK-265(Ex = 665 nm, Em = 725 nm, and Kd = 97 nM) to investigate the different stages of Aβ plaques in transgenic mice in vivo by using an Odyssey^®^ Infrared Imaging System (OIIS, LI-COR Biosciences, Lincoln, NE, USA). The results showed that the intensity of the NIRF signal was closely related to plaque load, indicating its practical value in monitoring directly the progression of Aβ aggregation, and opening the possibility of effective presymptom monitoring of Aβ deposition in the aging brain. In addition, Yang et al. [32] recently demonstrated that NIRF ocular imaging (NIRFOI), which was performed on IVIS^®^ Spectrum (PerkinElmer, Hopkinton, MA, USA) could provide higher sensitivity for Aβs than brain NIRF imaging does. By using NIRFOI with a near-infrared probe CRANAD-102 (Ex = 605 nm, and Em = 680 nm), they have observed significant CRANAD-102 expression in APP/PS1 at about 60 min, while WT has not. This high sensitivity is significant for both diagnosis and therapy monitoring.

## 5. Nonlinear Optical Microscopic Imaging

In linear optics, the electric polarization intensity induced by particles in the medium under the action of the external photoelectric field is proportional to the intensity of the incident photoelectric field, i.e., P=ε_0 χE. However, the electric polarization intensity in nonlinear optics is not proportional to the intensity of the incident of the photoelectric field. That is:$$P=\varepsilon_{0}\left(\chi^{\left(1\right)}E+\chi^{\left(2\right)}E^{2}+\chi^{\left(3\right)}E^{3}+\cdots \right)$$
where $\varepsilon_{0}$ is the permittivity of free space and $\chi^{\left(1\right)}$ is the linear susceptibility describing the linear optical process. $\chi^{\left(2\right)}$ and $\chi^{\left(3\right)}$ are the second- and third-order nonlinear optical susceptibilities.

Nonlinear optical microscopy can be categorized into one- and two-beam modalities. The one-beam modality includes multiphoton excited fluorescence (MPEF), second-harmonic generation (SHG), and third-harmonic generation (THG) microscopy. The two-beam modality includes coherent anti-Stokes Raman scattering (CARS), four-wave mixing, stimulated Raman scattering (SRS), and pump probe. Moreover, two-photon excited fluorescence (TPEF) provides a good molecular signal-to-noise ratio for images and has the ability of label-free morphological imaging. The strongest source of SHG is fibrous collagen. In addition, CARS is derived from molecular vibration. Thus, it has chemical selectivity and chemical specificity [42]. Nonlinear optical techniques have been widely used to study biological samples because of their high optical spatial and temporal resolution, the absence of required additional labeling, nondestructive manipulation, and chemical specificity. This part mainly introduces Aβ imaging by several nonlinear optical imaging techniques, including MPEF, SHG, THG, CARS, and SRS [42].

### 5.1. Multiphoton Excited Fluorescence Microscopy

The most common multiphoton imaging techniques are TPEF and three-photon excited fluorescence (3PEF) imaging. MPEF uses a near-infrared femtosecond laser pulse to replace the traditional ultraviolet light source, which has better imaging depth, lower optical damage, higher spatial resolution and contrast, and less photobleaching compared with the traditional single-photon FM [67,68,69,70]. In addition, MPEF has the advantage of selective imaging, which allows for the study of only the place of interest without being disturbed by the surrounding environment. Multiphoton microscopy has been widely used in biomedical fields in recent years and has shown great application potential in the diagnosis of a variety of diseases due to its advantages.

Previous studies have reported senile plaque imaging using MPEF, which can be considered as an ideal imaging tool for preclinical studies in AD mice models both in vivo and in vitro [71,72,73,74]. However, tissue diffusion limits the maximum depth of imaging of the mouse cerebral cortex with two-photon excitation FM. Either removal of the covered brain tissue or the introduction of an optical probe is required to solve this problem. MPEF has been employed for imaging amyloid plaques and their surrounding structures in the mouse brain in vivo by removing part of the skull or thinning skull together with different fluorescent labels such as ThS or thiazine red or fluorescently labeled anti-amyloid antibodies. Bacskai et al. [75] used the MPEF combined with ThS to directly observe the clearance of plaques in live mouse brains with immunotherapy. Christie et al. [71] used MPEF combined with ThS to observe Aβ plaques in vivo over several months. Chen et al. [76] successfully used a novel near-infrared probe for deep-brain imaging of amyloid plaques in vivo in AD mouse models without interference from a lipofuscin signal.

Based on previous studies, many researchers have focused on the factors affecting the dynamics of amyloid plaque formation and growth in vivo by using MPEF imaging. In 2003, the Alzheimer’s Disease Research Unit of Massachusetts General Hospital [71] reported the average number of plaques in the cortex of ThS-stained Aβ plaques from Tg2576 transgenic mice increased nearly sixfold from 12 to 22 months, while the size distribution of plaque diameters did not change significantly. Moreover, Yan et al. [77] found that the 6-month-old APP/PS1 mice showed more robust plaque growth than the 10-month-old mice, suggesting that plaque growth is more prominent in the early stage of disease and that early disease treatment may be more effective than treatment later in the disease.

Moreover, MPEF is especially useful for observing the relationship between Aβ deposits and microglial, amyloid angiopathy, and free radical production. Baik et al. [78] used live two-photon microscopy in vivo imaging and flow cytometry to confirm that the activated microglia surrounding Aβ plaques can phagocytose the plaques and die afterward. The accumulated Aβ is then released into the extracellular space by these dying microglia, which contributes to the growth of Aβ plaques. Cerebral amyloid angiopathy (CAA) is the most common AD complication, which is mainly manifested by the β-amyloid deposition in cerebrovascular walls. Moreover, Meyer-Luehmann et al. [79] developed a novel in vivo multiphoton imaging method and observed Aβ plaques and amyloid angiopathy by methoxy-X04 and thS, which was consistent with previous results obtained by Bacskai et al. Furthermore, Domnitz et al. [80] performed two-photon fluorescence imaging of the brain using a Ti:Sapphire laser at the 750 nm excitation wavelength, compared four different AD mouse models, and showed that Aβ accumulation in cerebral arteries begins as early as 9 months in Tg2576 mice. In addition, PS1xTg2576 and TgCRND8 mice also showed all superficial vessels affected by the end stage of disease, whereas aged PDAPP mice developed significant but not such extensive CAA. Additionally, Maria et al. [81] found that HS-84 and HS-169 were able to image Aβ aggregates and CAA by MPEF in the APP/PS1 mouse both in vivo and in vitro.

Since the near-infrared light used for multiphoton imaging does not oxidize the fluorogenic free radical indicators even after prolonged exposure, McLellan et al. [82] utilized the fluorogenic free radical indicator to mark Aβ plaques. Consequently, it turned out that the indicator was only associated with dense core Aβ plaques rather than diffuse plaques. In addition, the antioxidant therapy neutralizes these highly reactive molecules and may therefore be of therapeutic value in AD.

Hefendehl et al. reported using TPEF to study the formation of amyloid plaques in vivo in the brains of APP/PS1 transgenic mice over 6 months to evaluate the progression of amyloid plaques in vivo in the AD animal model [83]. A novel head fixation system was designed by this research group to provide stable and effective long-term tracking of individual plaques over time [83,84]. This head immobilization method allows for speedy relocation of previous imaged regions of interest (ROIs) inside the brain. This kind of ROIs can be automatically relocated and imaged from weeks to months with trivial rotational changes and only a few translational errors [84].

In addition to using a variety of fluorescent probes to detect Aβ, Kwan et al. [85] found the self-fluorescence of Aβ plaques and collagen in the tissue of AD mouse by MPEF and SHG images. Moreover, Wang et al. [86] successfully imaged the intracellular and extracellular Aβ deposition in the APP/PS1 mouse by label-free MPEF, which can prevent the cells from being damaged by some toxic fluorescent substances, providing the basis for advancing the use of label-free MPEF technology.

### 5.2. Second- and Third-Harmonic Generation Microscopy

SHG [87,88] and THG [89,90,91] microscopy imaging technologies have rapidly been developed in recent years. Second- and THG processes are different from multiphoton fluorescence excitation, and the harmonic wavelength generated is exactly equal to a half and one-third of the excitation wavelength. Moreover, no energy absorption was noted in the imaging process. Thus, damage to the sample could not be caused.

Despite the recent discovery that amyloid aggregates show increased multiphoton absorption properties, which are directly linked to fibrillization, most studies of amyloid structures were dependent on external probes [92,93,94] and only a few were label-free. The method to label-free image amyloid accumulates with MPEF microscopy had been proved very recently with an additional message obtained in the SHG imaging due to a high SHG susceptibility of the ordered fibrillar regions [43,92,94].

SHG and THG microscopies do not require sample labeling with exogenous probes, which could eliminate related toxic effects. These characteristics make the SHG an ideal tool for related applications, especially imaging of living cells and becoming a noninvasive imaging method [95,96]. However, SHG requires a noncentrosymmetric molecular structure, and only a very ordered structure can provide a sufficient SHG section. The SHG signal is not limited by the forward emission nature and can be collected backward. Thus, further development of SHG imaging is mainly reflected in the use of the combination with other optical imaging technologies. Moreover, SHG and TPEF are both nonlinear optical phenomena, which have many similarities, and are two completely different physical phenomena. The information obtained from the two phenomena can be compared with each other to confirm and supplement each other.

Kwan et al. [85] observed that the SHG signal of senile plaques in the hippocampus and brain vessels of transgenic AD mice corresponds with spontaneous fluorescence and ThS signals. Johansson et al. [94] used CLSM, MPEF, and SHG microscopy to perform label-free imaging of the amyloid plaques. Consequently, the subsequent two techniques have a high image contrast.

THG is a high harmonic technology developed after the second harmonic. The third harmonic is a third-order nonlinear effect, and the signal intensity is proportional to the third power of the excitation light. The second harmonic can only be generated in noncentrosymmetric materials, while the third harmonic does not have this requirement, which makes up for this defect in principle. By using a light source of 1263 nm, Chakraborty et al. [97] were able to see label-free Aβ in transgenic mouse brain slices with THG microscopy, consistent with the immunohistochemical results. The study provides a foundation for deep high-resolution THG imaging of brain Aβ in living mice. Furthermore, Chakraborty et al. had distinguished Aβ plaques, NFTs, and axons in brain sections [98] by comparing the additive-color multi-harmonic generation microscopy images in the cortex, striatum, and hippocampus of mice, which further advanced the application potential of unlabeled imaging in animal studies in vivo and in vitro.

### 5.3. Coherent Raman Scattering Microscopy

Raman spectroscopy, as a characterization method, is widely used in materials, physics, chemistry, biology, and other fields in recent years [99,100,101]. However, Raman scattering (RS) is very weak, and especially spontaneous RS. Its intensity is 8 to 10 orders of magnitude lower than that of fluorescence, which requires a much longer acquisition time, making RS imaging difficult to apply in biomedical research. Coherent RS (CRS) is a new microscopy imaging technology, which provides imaging contrast by detecting the characteristic vibration of the target molecule, and greatly enhances the RS signal based on nonlinear optical process, and improves the imaging rate and detection sensitivity. This section discusses the application of CRS in Aβ plaque imaging. Moreover, CRS microscopy mainly includes CARS [90,102,103,104] and SRS [105,106].

CARS imaging was introduced in 1999 by Xie’s group in the Pacific Northwest National Laboratory [104]. In 2008, the same group reported SRS [107]. Both CARS and SRS are third-order nonlinear optical processes in which the pump and Stokes light are used as excitation light. This means that these two processes happen simultaneously, but they are not generated and detected in the same way. Compared with CARS, the biggest advantage of SRS imaging is that no non-resonant background was noted, and its spectral profile is completely consistent with the spontaneous Raman spectrum.

CARS, which is nonlinear RS with anti-Stokes wave detection, provides an advanced noninvasive and label-free technique capable of selective imaging of major types of macromolecules (i.e., proteins, lipids, nucleic acids, and saccharides) [107]. Moreover, CARS imaging allows for label-free visualization of lipids at their characteristic frequency (2840 cm^−1^) [108,109,110], providing a tool to monitor lipid metabolism in live neuronal cells in real-time. CARS has a good lipid aggregation characterization, which is an important process that leads to Aβ aggregation. Observing the changes of lipids and Aβ is an important means to study AD. Using CARS microscopy, Kiskis et al. [44] have shown that a subpopulation of fibrillar Aβ plaques co-localizes with lipid deposits, and no lipid deposits were noted in diffuse plaques. Moreover, the lipid fluidity varies throughout the plaque region, which determines to what extent toxic Aβ oligomers could be released from the fibrillar plaques. This allows further investigation of the complex interactions between lipids and Aβ and deepens the understanding of the mechanisms of Aβ plaque toxicity.

Enejder et al. [111] found that two-photon FM of ThS staining did not provide a complete picture of the spatial distribution or molecular composition of Aβ plaques, which can be provided by CARS, promoting the extensive application of CARS microscopy in the field of AD research. Lee et al. [43] compared the microscopic images of CARS, TPEF, and SHG in normal AD mice. The result proved that the amount of lipids, Aβ, and collagens is greater in AD samples than in normal samples.

In addition to selectively imaging ordinary molecules based on their spectral differences, SRS can also see conformational changes in proteins. The formation of senile plaques is usually due to the accumulation of misfolded polypeptides which are the secondary structural changes of proteins from the α helix misfolded to β-sheet transitions. Moreover, the amide I band is highly sensitive to this kind of structural change. Ji et al. [112] applied the blue-shifted amide I band that happened in the forming of senile plaques to observe misfolded Aβ plaques in AD brain tissues via label-free multicolor SRS imaging.

## 6. Summary and Outlook

Optical imaging can achieve both labeled and unlabeled imaging. Moreover, the former has good specificity and the latter is suitable for living organisms. The development of optical imaging has experienced a process from qualitative to quantitative, two-dimensional to three-dimensional, in vitro to in vivo imaging. Based on the advantages of noninvasiveness, small biological damage, high sensitivity, low cost, fast imaging speed, and three-dimensional imaging, optical imaging has been widely used in the detection of biological processes at the cellular or molecular level. Since AD cannot be detected at an early stage and cannot be effectively cured, early detection of Aβ has very important research significance and application value for the treatment and prevention of AD. This review briefly introduced the principles of commonly used optical imaging methods and their applications in Aβ plaque imaging. To provide a reference in this field, this review summarized the various optical imaging methods that have been used in the current Aβ study as shown in Table 1, and the characteristics of these optical imaging methods are summarized in Table 2.

Optical imaging has demonstrated promising results in the study of Aβ plaques. However, some problems that need to be resolved still exist before their clinical application. Introducing multimodal nonlinear optical imaging into biomedical research can overcome this limitation because every nonlinear optical microscopy imaging modality is only sensitive to specific molecules or structures. Moreover, Gualda et al. [113] used the combination of CARS and TPEF to study the interaction between peptides and lipid bilayers. Using a multimodal multiphoton system, Lee et al. [43] obtained and compared the CARS, TPEF, and SHG microscopy images of CA-1 and DG regions of normal and AD mice, showing that the multimodal microscopy images can reveal distinct molecular structures and components (lipids, amyloid-beta fiber, and collagen) of brain tissue associated with AD. New active focusing techniques are introduced, involving fiber optics and endoscopes, to improve the optical imaging depth. Choi et al. [114] developed a reflective matrix microscope that allowed them to perform microscopy mapping of neural networks in the brain tissue through the intact mouse skull, without spatial resolution loss. In terms of overcoming spatial resolution, better imaging quality can be obtained by increasing the aperture of the system and adopting a shorter working wavelength. In the aspect of improving the detection sensitivity, it may be an effective solution to reduce and eliminate the noise from the modulation of the light source and the data acquisition design. Moreover, the ghost imaging method can be used to accelerate the imaging speed of a super-resolution fluorescent optical microscope, which is expected to capture the biological processes occurring in cells at a submillisecond speed. Thus, optical imaging technology can be predicted to have more progress in AD application with the unremitting efforts of researchers.

## Figures and Tables

**Figure 1 biosensors-11-00255-f001:**
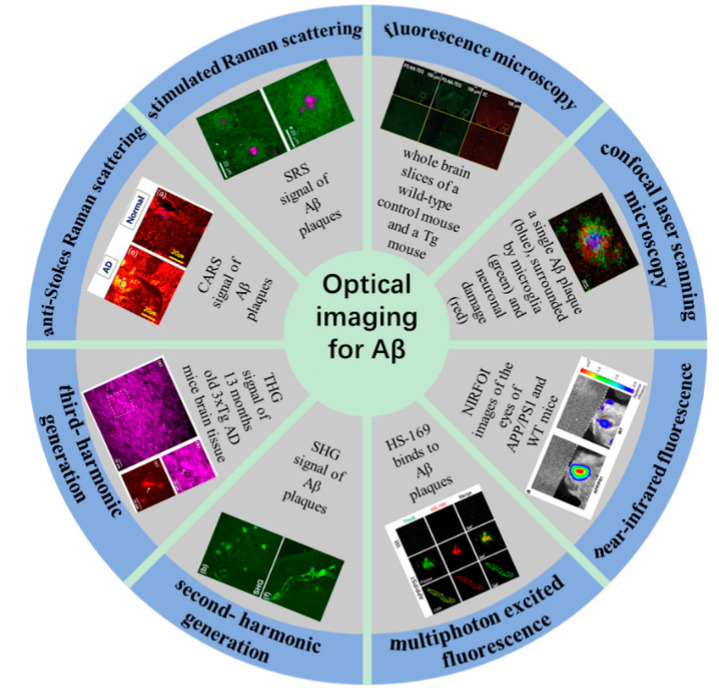
Optical imaging for Aβ plaque detection and visualization.

**Figure 2 biosensors-11-00255-f002:**
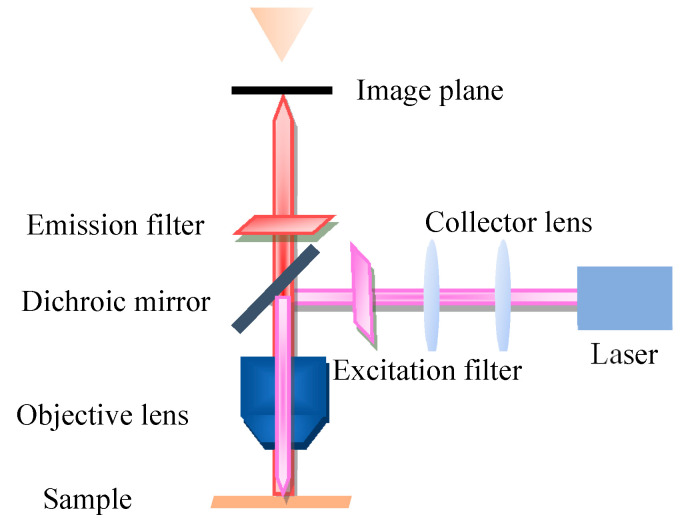
Principle scheme of fluorescence microscopy.

**Figure 3 biosensors-11-00255-f003:**
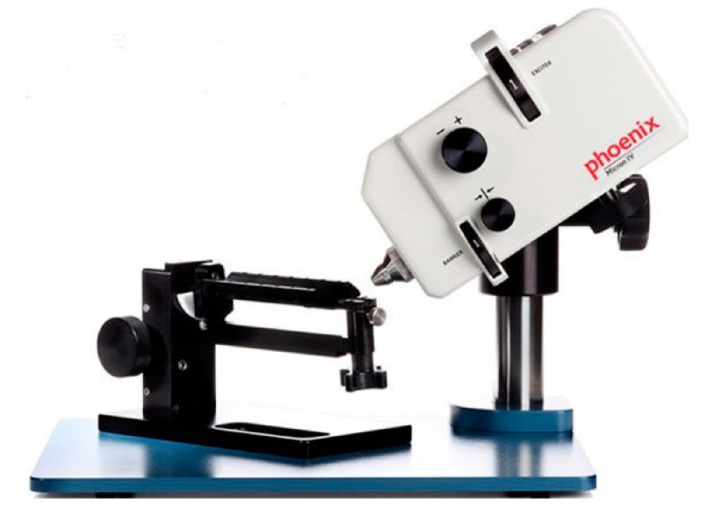
Physical photos of a Micron II rodent retinal imaging microscope.

**Figure 4 biosensors-11-00255-f004:**
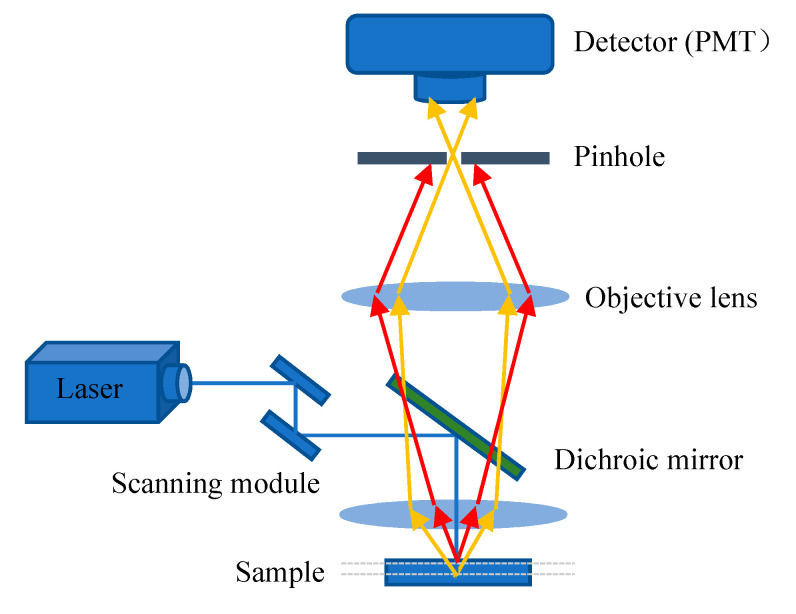
Principle scheme of confocal laser scanning microscopy.

**Figure 5 biosensors-11-00255-f005:**
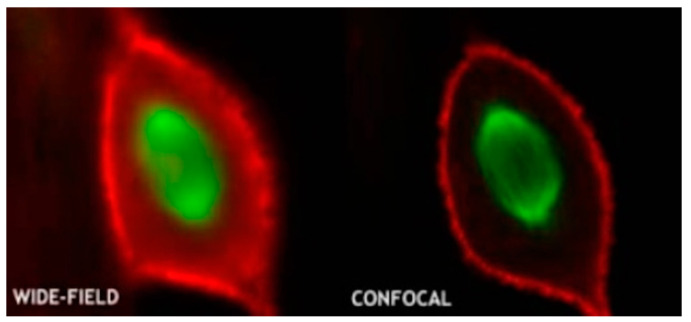
Conventional microscopy vs. confocal microscopy (reproduced with permission from the National Institute of Biological Sciences, Beijing).

**Table 1 biosensors-11-00255-t001:** Optical imaging in Aβ research.

Optical Imaging Method	Probes	Parameters	Imaged Samples	Reference
FM	curcumin	λex = 550/25 nm λem = 605/70 nm Resolution: 0.25 μm	retina slices	[45]
FM	Cy5, CRANAD-2	λex = 649 nm, λex = 649 nm λem = 675 nm, λem = 715 nm Resolution: 5 μm laser power: 5~50 mW	agar phantom	[48]
FM CLSM	PiB flutemetamol	λex = 330~390 nm λem = 390~450 nm Resolution: 0.10 × 0.108 × 0.11 μm	brain slices	[49]
FM super-resolution images	PD-NA, PD-NA-TEG	λex = 405 nm λem = 500~546 nm Resolution: sub-100 nm laser power: 50 mW	brain slices	[58]
LMI-FM	HS-169	λex = 532 nm Resolution: 20 μm	In vivo brain	[59]
fMOST	DANIR-8c	Resolution: 0.32 × 0.32 × 2 μm	In vitro brain	[60]
CLSM	ThT	λex = 450 nm λem = 482 nm	oAβ_42_	[51]
CLSM	ThS	1024 × 1024 pixel	brain slices	[52]
CLSM	12F4	1024 × 1024 pixel	brain slices	[55]
CLSM	specific monoclonal M78		pheochromocytoma	[56]
CLSM	curcumin micelles 12F4	λex = 405 nm λem = 525 nm a laser beam (2 mW) at 514 nm for 6 min.	brain and retinal slices	[57]
NIRF	AOI987, NIAD-11, NIAD-16	λex = 650 nm, λem = 670 nm λex = 545 nm, λem = 690 nm λex = 470 nm, λem = 720 nm	brain slices	[25,65]
NIRF	CRANAD-2	λex = 640 nm, λem = 805 nm laser power: 10 mW/cm^2^ 532 × 256 pixels	In vivo and in vitro brain	[64]
NIRF	THK-265	λex = 665 nm, λem = 725 nm 169 or 84 μm resolution	brain slices	[66]
NIRF	CRANAD-102	λex = 605 nm, λem = 680 nm	brain slices	[32]
MPEF	ThS	two-photon fluorescence λex = 750 nm λem = 380~480 nm laser power after the objective: 10 mW, pulse 60–100 fs Resolution: 1 μm depth = 150 μm	In vivo brain	[71]
MPEF	methoxy-X04	two-photon fluorescence λex = 750 nm λem = 435~485 nm depth = 200 μm	In vivo brain	[77]
MPEF	methoxy-X04	two-photon fluorescence λex = 850 nm λem = 460 nm laser power < 35 mW	In vivo brain	[78]
MPEF	methoxy-X04	two-photon fluorescence λex = 800 nm λem = 380~480 nm Resolution: 150 × 150 × 1 μm	In vivo brain	[79]
MPEF	ThS	two-photon fluorescence λex = 750 nm λem = 380~480 nm depth = 200 μm Resolution: 615 × 615 μm	In vitro brain	[80]
MPEF	HS-84, HS-169	λex = ~375 nm and ~535 nm (double excitation peaks), λem = ~ 665 nm resolution of 512 × 512 pixels depth = ~200 μm	brain slices	[81]
MPEF	ThS	two-photon fluorescence λex = 750 or 800 nm λem = 380~480 nm depth = 200 μm Resolution: 615 × 615 μm	In vivo brain	[82]
MPEF, SHG	Label-free	two-photon fluorescence λex = 810 nm SHG signals λem = 395~415 nm TPEF signals λem = 430~690 nm laser power: 5~10 mW 1024 × 1024 pixel	brain slices	[85]
MPEF, SHG	Label-free	MPEF λex = 830 nm SHG signals λem = 387 nm TPEF signals λem = 400~550 nm laser power: 25 mW	brain slices	[86]
CLSM, MPEF, SHG	Label-free	CLSM λex = 405 nm, λem > 420 nm MPEF λex = 910 nm SHG signals λem = 420~460 nm TPEF signals λem = 495~540 nm laser power: 680 mW pixel sizes < 200 nm	brain slices	[94]
THG	Label-free	MPEF λex = 1262 nm, λem > 430 nm laser power: 20 mW 1024 × 1024 pixels	brain slices	[97,98]
CARS	ThS Cy2	Stokes λex = 1064 nm Pump λex = 817 nm the CH2 stretch vibration: 2845 cm^−1^ ThS signal: short-pass filters (600SP and 2 × 750SP, Ealing) Cy2 signal: band-pass filter (525/50 nm, Chroma) average laser power: 25 mW	brain slices	[44,111]
SRS	ThS	Stokes λex = 1064 nm Pump λex = 720~990 nm maximum brightness of the plaque images: 1670 cm^−1^ the CH2 stretch vibration at 2845 cm^−1^ resolution: ~8 cm^−1^	brain slices	[112]
CARS TPEF SHG	Label-free	Stokes λex = 1064 nm Pump λex = 800 nm CARS: HQ650/20 m, Chroma, TPEF: FF01-550/88, SHG: FF01-390/18, Semrock resolution: ~5 cm^−1^ laser power1: 20 mW laser power2: 3 mW	brain slices	[43]

In vivo brain: the brain of a living Alzheimer’s mouse. In vitro brain: the intact, postmortem brain of Alzheimer’s mouse.

**Table 2 biosensors-11-00255-t002:** Characteristics of optical imaging methods and their applications in biology.

Optical Imaging Method	Advantages	Disadvantage	Applications in Biology
FM	Easy to operate, low cost	Low resolution and low contrast	Thin biological samples, slice
CLSM	High resolution, high contrast	Expensive, damage to living cells, time-consuming	Thick biological samples
NIRF	Fast imaging speed, high penetration, non-destructive,	Poor sensitivity, vulnerable to interference	In vivo imaging
MPEF	High penetration depth, low phototoxicity	High cost, complex system	In vivo imaging
SHG	No photobleaching, label-free	The signal is weak and difficult to collect	Occurs only in an asymmetric medium (e.g., collagen)
THG	No photobleaching, label-free	The signal is weak and difficult to collect	Can occur in any medium (whether symmetric or not)
CARS	Good chemical specificity, small light damage, high sensitivity, high spatial resolution, fast scanning speed	Strong non-resonant background	In vivo imaging
SRS	Low background noise, fast scanning speed	Expensive, complex system	In vivo imaging

## Data Availability

Not applicable.

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
