# Peer review of "Optical Imaging of Beta-Amyloid Plaques in Alzheimer’s Disease"

_biosensors, 2021, doi:10.3390/bios11080255_

Round 1

Reviewer 1 Report

The manuscript reviews recent works about optical imaging of Abeta plaques in Alzheimer’s disease (AD). Existing imaging techniques for the clinical AD diagnosis are discussed. Then, the optical imaging techniques applied to image Abeta plaques in AD is described in detail, including fluorescence microscopy (FM), confocal laser scanning microscopy (CLSM), near-infrared fluorescence imaging (NIRF), and nonlinear optical microscopy. Overall, the manuscript is well written. A few issues should be addressed to improve the manuscript before it can be published.

  1. The title of section 2 may be too general. FM may include conventional FM (using a CCD camera, for example) and confocal FM (the case for section 3). Besides, the title of section 4, NIRF, can be categorized as a kind of FM. Therefore, the authors may consider rename the titles of sections 2-4 in a more appropriate and precise way to avoid ambiguity. For example, change the title of section 2 to “Conventional Fluorescence Microscopy.”
  2. In Table 1, imaged samples are summarized. There are “retina sections”, “brain slices”, “brain sections”, “in vivo brain”, “brain”, etc. This is confusing. Specifically, (1) are “brain slices” the same as “brain sections”?, (2) what does it mean by “brain” (“in vivo brain”?, “in vitro brain”?, “brain sections”?)? It would be better to use the same phrase to describe the same content. Further explanation about a certain type of the imaged samples, if necessary, at the end of the table may also be helpful. Besides, are all the samples from animals? It would be useful for the readers if the authors can further specify which animal (e.g., a mouse, a rat, etc.) was used to obtain the samples.
  3. In all these optical techniques (including FM, CLSM, NIRF, MPEF, SHG, THG, CRS), it would be useful to specify which techniques are label-free, which requires probes, and which can be used in both ways (label-free and probes), if any. This can be done by another table or simply described in text.
  4. A few typos: (1) “plague” in pages 3 and 4, which should be plaque; (2) “noninvasive” in page 15, which should be noninvasiveness.

Author Response

Response 1: We agree with the reviewer. Section 2 described the use of conventional fluorescence microscopy. Section 3 described the use of confocal fluorescence microscopy, and section 4 describes the use of near-infrared fluorescence imaging (NIRF) technology. According to reviewer’s suggestion, we have changed the title of section 2 to “Conventional Fluorescence Microscopy.”

Response 2: Thank you very much for the suggestions! We have rewritten and completed the information in Table 1. “brain slices” is same as “brain sections” and we have used the same phrase to describe the same content. In vivo brain means the brain of a living Alzheimer's mouse, in vitro brain means the intact, postmortem brain of Alzheimer's mouse. This has been clarified in the manuscript. The revised text is highlighted with red.

Response 3: In general, probes are needed for FM, CLSM, NIRF and MPEF, while SHG, THG and CRS can achieve label-free imaging. However, due to the weak auto-fluorescence signal of Aβ, some study can successfully observe the Aβ by label-free MPEF, for example, Wang et al. photographed intracellular and extracellular β deposition on APP/PS1 mouse by label-free MPEF. [Wang, S., B. Lin, G. Lin, C. Sun, R. Lin, J. Huang, J. Tao, X. Wang, Y. Wu, L. Chen, et al. Label-free multiphoton imaging of beta-amyloid plaques in Alzheimer's disease mouse models. Neurophotonics. 2019 6(4): 045008.] In addition, probes can be used to enhance the signal of CARS for samples with weak self-fluorescence signal. [Kiskis, J., H. Fink, L. Nyberg, J. Thyr, J. Y. Li and A. Enejder. Plaque-associated lipids in Alzheimer's diseased brain tissue visualized by nonlinear microscopy. Sci Rep. 2015 5: 13489.]

Response 4: Thank you for the suggestion! We have corrected the typos in the article.

Reviewer 2 Report

In the work of Ziyi Luo, HX et al. titled Optical Imaging of Beta-Amyloid Plaques in Alzheimer's Disease is an important issue for the scientific community, where imaging techniques are important tools in evaluating findings related to AD. But I think that important points in the manuscript need to be improved: - I believe that for a comparative assessment of studies in the literature on each technique, it is recommended to create tables of studies indicating important aspects, so that the reader could take advantage of more information from the articles on the subject. In this table technical data can be placed, but what I think is most important is to provide information in a summarized form showing the findings, results, etc., where the optical image had its contribution. - Basic optical concepts I consider unnecessary to include in the article. - In the NIRF technique, I believe that the problems that this technique presents with autofluorescence, food, type of animal, signal attenuation etc. could be deepened, but there are other factors that interfere with image acquisition. This additional information would be of great importance, showing graphs, etc. It is not just a matter of quoting, but a little deeper into these factors.

Author Response

Response 1: We thank the reviewer very much for his comments and suggestions! We have deleted some schematic figures and shortened the length of the basic optical concepts. To provide a reference in this field, various optical imaging methods that have been used in the current Aβ study are summarized and shown in Table 1(in the manuscript), and their characteristics are summarized in Table 2. We also provide more discussions on NIRF imaging in the manuscript, as shown in the following.

Table 2. Characteristics of optical imaging methods and their applications in biology

Optical imaging method

Advantages

Disadvantage

Applications in biology

CFM

Easy to operate, low cost

Low resolution and low contrast

Thin biological samples, slice

CLSM

High resolution, high contrast

Expensive, damage to living cells, time-consuming

Thick biological samples

NIRF

Fast imaging speed, high penetration, non-destructive, 

poor sensitivity, vulnerable to interference

In vivo

MPEF

High penetration depth ~8mm, low phototoxicity

High cost

The complex technology

In vivo imaging

SHG

No photobleaching, label-free

The signal is weak and difficult to collect

Occurs only in an asymmetric medium (e.g., collagen)

THG

No photobleaching, label-free

The signal is weak and difficult to collect

Can occur in any medium (whether symmetric or not)

CARS

Good chemical specificity, small light damage, high sensitivity, high spatial resolution, fast scanning speed

Strong non-resonant background

In vivo imaging

SRS

Low background noise, fast scanning speed

Expensive, complex system,

In vivo imaging

Near infrared fluorescence imaging has many advantages, such as higher penetration depth, high sensitivity, noninvasiveness, and low cost. Therefore, the small molecule probe used in near infrared fluorescence imaging has become a hot research field. The design and optimization of molecular probe is an important step in the study of molecular imaging, and it is the basis for the efficient and specific diagnosis of diseases. In vivo quantitative evaluation of the neuropathological process of AD by the design of highly efficient and specific molecular probes plays an important role in both basic and clinical studies of AD. NIRF fluorescent imaging mainly includes the development of biocompatible NIR fluorescent dyes and the synthesis of various probes. The ideal NIRF fluorescent probe should have the following characteristics: (1) targeting Aβ with high selectivity and high affinity; (2) the emission wavelength is in the near infrared spectral range; (3) rapid penetration of the blood-brain barrier (BBB) and rapid clearance in the organism; (4) high quantum yield; (5) low affinity with bovine serum albumin (BSA); (6) easy synthesis, and most importantly, (7) binding to Aβ plaques, which should significantly change its fluorescence properties (e.g., fluorescence intensity, fluorescence lifetime, emission wavelength, and quantum yield). At present, it is difficult for any probe to fully meet these conditions. Therefore, the development of new near-infrared probe to reduce the effect of endogenous fluorescence, light scattering and light absorption of biological tissues will be of great significance for the applications of NIRF imaging in AD animal models.

Reviewer 3 Report

This review paper has summarized different optical imaging techniques for diagnosing Alzheimer's disease biomarker Aβ plaques. I believe there are some issues the author needs to address.

  1. The author has spent too much space explaining the theory of each technique. Although it is very friendly to readers without much background, it also hurts the depth of the paper. Especially, the author has many schematic figures to explain the microscope and energy level diagram.
  2. Following the first issue, the author should add more figures from the referred paper. Such as the microscope image of different samples or comparison image before and after using a novel technique.
  3. Page 3, line 102. Please explain why does the plague exhibits weak endogenous fluorescence although Aβ itself does not fluoresce?
  4. Page 4, line 109. Here the author has introduced an in-vivo fluorescence microscope. It is better to add a figure to explain how do they design it. Actually, in vivo fluorescence microscope is very challenging. The author should focus on their design.
  5. Page 7, line 227. What is CRANAD-2? Please provide the full name for this abbreviation. 
  6. Subtitle 5.3. The major drawback of Raman spectroscopy is its poor repeatability, which makes it very difficult to analyze the analyte quantitatively. Please explain how do these referred papers solve this issue? 
  7. The distribution of Aβ plaques is not homogeneous. It's mainly located at the cerebellar cortex. It is necessary to explain the rationality to detect Aβ plaques on the retinal.

Author Response

Response 1: We thank the reviewer very much for this comment and suggestion! We have deleted some schematic figures and shortened the length of the theory of various techniques to ensure the depth of the paper.

Response 2: According to the reviewer’s suggestion, in the introduction we have designed Figure 1 to illustrate various optical imaging techniques for Aβ plagues detection and visualization. Figure 4 was provided to compare conventional microscopy with confocal microscopy.

Response 3: We tried to explain that although Aβ protein itself does not have strong endogenous fluorescence, the specific luminescence mechanism of fluorescent probe can be used to realize the visualization of Aβ protein. Since this sentence did not clearly express our meaning, we have modified this sentence in the manuscript to make it clear and understandable.

Response 4: In the research of Karonyo-Hamaoui et al. in vivo imaging of the retina was performed in live mice utilizing Micron II rodent retinal imaging microscope, which is a comprehensive ophthalmology research platform developed by Phoenix Research Labs specially designed for small experimental animals' eyes. It can provide bright field and fluorescence images of retina of rats and mice, and fluorescein fundus angiography. The accuracy of retinal imaging is ≤4um in mice and ≤8um in rats. The physical photos of the system as shown in Figure 3.

Response 5: CRANAD-2 (not abbreviation), a compound derived from curcumin, has a maximum absorption/emission wavelength of 640 nm/805 nm. When combined with Aβ, it exhibits obvious optical properties such as a 70-fold increase in fluorescence intensity, Stokes shift (from 715 nm to 805 nm), and a 67-fold increase in quantum yield. The probe has a high affinity with Aβ polymer (Ka38.69 nmol/L), reasonable lipophilicity (logP=3.0), stable structure and weak interaction with albumin. [Nesterov, E. E., J. Skoch, B. T. Hyman, W. E. Klunk, B. J. Bacskai and T. M. Swager. In vivo optical imaging of amyloid aggregates in brain: design of fluorescent markers. Angew Chem Int Ed Engl. 2005 44(34): 5452-5456.]

Response 6: In the referred papers, because the spectral shift of amide I band is around 10 cm−1, it requires a Coherent Raman Scattering (CRS) Microscopy with sufficient spectral resolution to differentiate the two Raman bands. At this point, picosecond narrow band laser–based CRS has the best spectral resolutions, while femtosecond laser–based methods usually have resolutions worse than 20 cm−1 and hence may not be suited for these studies. And spectra of different components will serve as the basis for quantitative numerical decomposition for the later spectral imaging data analysis based on linear algebra. Besides, the visualization of morphological differences in protein and lipid distributions in Alzheimer´s plaques were compared by using CARS and two-photon fluorescence (TPF) microscopy to demonstrate its reliability. SRS and antibody staining results on the same tissue section were also compared from frozen AD mouse brain section to fresh AD mouse brain. These results strongly support that CRS microscopy could indeed image Aβ.

Response 7: For the study of Aβ plaques on the retinal, Karonyo-Hamaoui et al. reported in 2011 that Aβ plaques can be found earlier in the retina than that in the brain and it accumulates as the disease progresses. Koronyo-Hamaoui's team and other researchers found Aβ deposits in mouse models and in autopsy of the retinas of patients with AD. They used curcumin as a fluorescent label to perform in vivo fluorescence imaging of Aβ plaques in the retina of AD-Tg mice, and those plaques were not detected in the non-Tg mice. Subsequently a number of studies have shown that Aβ is present in the retina.

[Koronyo-Hamaoui, M., Y. Koronyo, A. V. Ljubimov, C. A. Miller, M. K. Ko, K. L. Black, M. Schwartz and D. L. Farkas. Identification of amyloid plaques in retinas from Alzheimer's patients and noninvasive in vivo optical imaging of retinal plaques in a mouse model. Neuroimage. 2011 54 Suppl 1: S204-217.

Ziv, Y., Avidan, H., Pluchino, S., Martino, G., Schwartz, M., . Synergy between immune cells and adult neural stem/progenitor cells promotes functional recovery from spinal cord injury. Proc. Natl. Acad. Sci. U. S. A. 2006 103: 13174–13179.

Frost, S., Y. Kanagasingam, H. Sohrabi, J. Vignarajan, P. Bourgeat, O. Salvado, V. Villemagne, C. C. Rowe, S. L. Macaulay, C. Szoeke, et al. Retinal vascular biomarkers for early detection and monitoring of Alzheimer's disease. Transl Psychiatry. 2013 3: e233.

Tes, D., K. Kratkiewicz, A. Aber, L. Horton, M. Zafar, N. Arafat, A. Fatima and M. R. Avanaki. Development and Optimization of a Fluorescent Imaging System to Detect Amyloid-beta Proteins: Phantom Study. Biomed Eng Comput Biol. 2018 9: 1179597218781081.]

Round 2

Reviewer 2 Report

All recommendations requested for the authors were carried out.

Reviewer 3 Report

The author has included much more information in the revised manuscript. I have no further questions. Congratulations!